# Searching for the Antioxidant, Anti-Inflammatory, and Neuroprotective Potential of Natural Food and Nutritional Supplements for Ocular Health in the Mediterranean Population

**DOI:** 10.3390/foods10061231

**Published:** 2021-05-28

**Authors:** Mar Valero-Vello, Cristina Peris-Martínez, José J. García-Medina, Silvia M. Sanz-González, Ana I. Ramírez, José A. Fernández-Albarral, David Galarreta-Mira, Vicente Zanón-Moreno, Ricardo P. Casaroli-Marano, María D. Pinazo-Duran

**Affiliations:** 1Ophthalmic Research Unit “Santiago Grisolía” Foundation for the Promotion of Health and Biomedical Research of Valencia FISABIO, 46017 Valencia, Spain; vavema@alumni.uv.es (M.V.-V.); jj.garciamedina@um.es (J.J.G.-M.); silsanz5@ext.uv.es (S.M.S.-G.); dolores.pinazo@uv.es (M.D.P.-D.); 2Ophthalmic Medical Center (FOM), Foundation for the Promotion of Health and Biomedical Research of Valencia (FISABIO), 46015 Valencia, Spain; cristinaperismartinez0@gmail.com; 3Department of Surgery, University of Valencia, 46019 Valencia, Spain; 4Spanish Net of Ophthalmic Research “OFTARED” RD16/0008/0022, Institute of Health Carlos III, 28029 Madrid, Spain; airamirez@med.ucm.es (A.I.R.); dgalarreta@saludcastillayleon.es (D.G.-M.); rcasaroli@ub.edu (R.P.C.-M.); 5Department of Ophthalmology, General University Hospital “Morales Meseguer”, 30007 Murcia, Spain; 6Department of Ophthalmology and Optometry, University of Murcia, 30120 Murcia, Spain; 7Cellular and Molecular Ophthalmobiology Group, Department of Surgery, Faculty of Medicine and Odontology, University of Valencia, 46010 Valencia, Spain; 8Department of Immunology, Ophthalmology and Otorrinolaringology, Institute of Ophthalmic Research “Ramón Castroviejo”, Complutense University of Madrid, 28040 Madrid, Spain; joseaf08@ucm.es; 9Department of Ophthalmology. University Clinic Hospital of Valladolid, 47003 Valladolid, Spain; 10Faculty of Health Sciences, International University of Valencia, 46002 Valencia, Spain; 11Departament of Surgery, School of Medicine and Health Sciences, Clinic Hospital of Barcelona, Universitat de Barcelona, 08036 Barcelona, Spain

**Keywords:** eye diseases, natural food, antioxidant/anti-inflammatory/neuroprotective properties, broccoli, saffron, tigernuts-*chufa de Valencia*, walnuts, Mediterranean diet, nutritional supplements

## Abstract

Adherence to a healthy diet offers a valuable intervention to compete against the increasing cases of ocular diseases worldwide, such as dry eye disorders, myopia progression, cataracts, glaucoma, diabetic retinopathy, or age macular degeneration. Certain amounts of micronutrients must be daily provided for proper functioning of the visual system, such as vitamins, carotenoids, trace metals and omega-3 fatty acids. Among natural foods, the following have to be considered for boosting eye/vision health: fish, meat, eggs, nuts, legumes, citrus fruits, nuts, leafy green vegetables, orange-colored fruits/vegetables, olives-olive oil, and dairy products. Nutritional supplements have received much attention as potential tools for managing chronic-degenerative ocular diseases. A systematic search of PubMed, Web of Science, hand-searched publications and historical archives were performed by the professionals involved in this study, to include peer-reviewed articles in which natural food, nutrient content, and its potential relationship with ocular health. Five ophthalmologists and two researchers collected the characteristics, quality and suitability of the above studies. Finally, 177 publications from 1983 to 2021 were enclosed, mainly related to natural food, Mediterranean diet (MedDiet) and nutraceutic supplementation. For the first time, original studies with broccoli and tigernut (*chufa de Valencia*) regarding the ocular surface dysfunction, macular degeneration, diabetic retinopathy and glaucoma were enclosed. These can add value to the diet, counteract nutritional defects, and help in the early stages, as well as in the course of ophthalmic pathologies. The main purpose of this review, enclosed in the Special Issue “Health Benefits and Nutritional Quality of Fruits, Nuts and Vegetables,” is to identify directions for further research on the role of diet and nutrition in the eyes and vision, and the potential antioxidant, anti-inflammatory and neuroprotective effects of natural food (broccoli, saffron, tigernuts and walnuts), the Mediterranean Diet, and nutraceutic supplements that may supply a promising and highly affordable scenario for patients at risk of vision loss. This review work was designed and carried out by a multidisciplinary group involved in ophthalmology and ophthalmic research and especially in nutritional ophthalmology.

## 1. Introduction

The World Health Organization (WHO) has recently confirmed that up to 1 billion people worldwide suffer from visual dysfunction that can be prevented or treated to avoid blindness [1]. According to global population growth indices, the number of individuals with mild-to-moderate visual impairment as well as those with severe vision loss and blindness, are worryingly increasing everywhere. It has been recently reported that most challenging ocular disorders are uncorrected/under-corrected refractive errors, ocular surface dysfunction (OSD)/dry eye disease (DED), cataracts, glaucoma, diabetic retinopathy (DR) and age-related macular degeneration (AMD). In fact, severe visual impairment and blindness due to cataract or refractive error constitutes half of all global cases, being glaucoma the most common cause of irreversible blindness [1,2,3], with DR as the first cause of visual disability in working-age adults [1,2,3,4] and the AMD constituting the first cause of blindness in the elderly [1,2,3,4,5]. Overall, the above disorders seriously impair the quality of life related to vision. It is a priority to better understand the pathogenic mechanisms of the most challenging ophthalmic diseases and systemic disorders with eye involvement [6], to develop new diagnostic and therapeutic strategies from both the pre-clinical and clinical areas of concern.

## 2. State of the Art

Inflammation/immune response and neurodegeneration processes are common pathogenic mechanisms associated with the most prevalent ocular diseases [1,2,3,4,5,6] The cellular and molecular mechanisms underlying these shifts display interesting similarities.

Disbalance between the generation of reactive oxygen and nitrogen species (ROS and RNS) and the activation of antioxidant defenses is known as oxidative stress, a challenging mechanism involved in a wide variety of ocular disorders [7,8]. In fact, the eyes are exposed to environmental and endogenous agents which makes them especially sensitive to oxidative injury by ROS [superoxide anions (·O¯), hydrogen peroxides (H_2_O_2_), the most damaging hydroxyl free radical (·OH)] and/or by RNS [nitric oxide radical (NO·), peroxynitrite (ONOO^−^), nitrogen dioxide radical (NO_2_·)

Associated diseases to oxidative and nitrosative stress and its downstream effectors, much of these age-related processes, are prevalent pathologies potentially leading to blindness [6,7,8,9]. Active ROS present a dual role, acting as both destructive and constructive species. Thus, they participate in many activities for the preservation of cellular homeostasis, but in high concentrations, they lead to a situation of oxidative stress involved in the damage of cellular structures [10,11]. Additionally, antioxidants can act as classical defenses, and as sensors of intracellular oxidants and regulators of the redox signaling [10,11]. It has widely been reported that ROS and RNS are involved in signal transduction pathways of proliferation and differentiation, inflammation and immune response, angiogenesis, metabolic dysfunction, and neurodegeneration processes, in the eyes [7,8,9,12,13,14,15,16,17].

Cellular immune response depends on the T lymphocytes, representing 70% of all population, along with the T-cell surface receptor (TCR) the responsible of fragments of antigen recognition (as peptides bound to molecules of the major histocompatibiligy complex). T-cells and TCRs accomplish the following functions: cytokine production, helper-T-lymphocytes, immunosuppression control, and unpleasant target cell destruction [18]. In addition to the T-cells and TCRs signaling pathways, several co-stimulatory and co-inhibitory molecules, known as immune checkpoints (ICs), regulate the T-cell activities. Under normal conditions, these molecules are essential in maintaining immune homeostasis and preventing autoimmunity by controlling the type, extent, and time span of the immune responses [19]. There are numerous different T cell subtypes having their own specific identifiable surface markers and displaying different roles. Two major T lymphocyte types can be identified according to the presence of the CD4 and CD8 cell surface molecules, and subsequently classified into the Th1 cells and Th2 cells, also producing the correspondent cytokines [9]. A wide variety of pro-inflammatory cytokines have been reported in humans and animal models, with an increasing interest in relation to ophthalmic diseases, over recent years [20,21,22,23,24]. The microglia of the central nervous system protect the organs and tissues under normal conditions, by responding swiftly to the injury signals. However, these activities are reverted in chronic neurodegenerative diseases, where the activated microglia are shifted on a pro-inflammatory phenotype that, in turn, release pro-inflammatory cytokines and different neurotoxic substances such as ROS and RNS, proteolytic enzymes, specific neurotransmitters, and others [12,15,16,17,22,24].

Neurodegeneration refers to the progressive damage and loss of both the structure and function of nerve cells and the vascular system, occurring in specific sites of the central and peripheral nervous system and the neurosensory organs [15]. Retinal and optic nerve neurodegenerative diseases, the majority of them incurable, are increasing worldwide, representing a serious health problem and a big financial burden for the countries [6,12,15,16,17,24,25,26,27], among them, the multifactorial sight-threatening retinal and optic nerve diseases: AMD, RD, glaucoma, retinitis pigmentosa, macular dystrophies, uveoretinitis, pathologic myopia, retinal vascular occlusion, as well as the optic neuropathies [6,12,15,16,17,24,25,26,27,28,29,30,31,32].

Nutritional intervention has been evoked as one important tool for protecting the eyes and vision. Natural food and its derivatives play essential roles in health and well-being, and it has been characteristically associated with defense mechanisms against microbial agents, xenobiotics and/or physical factors, also demonstrating an immense range of health-promoting perspectives for achieving a pleasant lifespan [33,34]. Large population studies on the role of nutrition in eye health have reported controversial results over decades, mainly regarding cataracts, AMD, RD, OSD/DED, and glaucoma, and also slowing progression of the above sight-threatening diseases, by dietary interventions has also been discussed [6,8,35,36,37,38,39,40,41,42,43,44,45,46,47,48,49,50,51,52,53,54].

It has recently shown a reduced global risk of DR in a Mediterranean population with type 2 diabetes mellitus (T2DM), according to clinical, biochemical and lifestyle biomarkers including the adherence to the Mediterranean Diet (MedDiet) [40,55], and also a reduction in the enlargement of drusen, the most relevant manifestation of AMD, by the MedDiet and lifestyle incorporating fruits, vegetables, legumes and fish [56]. In this concern, the randomized clinical trial “Prevención con Dieta Mediterránea” (PREDIMED) demonstrated a reduced risk of DR in middle-aged/older individuals with type 2 diabetes, with the intake of at least 500 mg/d of dietary long-chain omega 3 polyunsaturated fatty acids (ω3 PUFAs), by means of two weekly servings of oily fish [57]. Additionally, the PREDIMED group showed that intake of skimmed yogurt was associated with lower risk of cataracts in the elderly Mediterranean population with high cardiovascular risk [58].

Nutrition clearly makes a difference to eye health and vision care [54]. In fact, coordinated, multidisciplinary interventions are essential to deal with the role of natural food, and nutritional supplements to achieve better knowledge of the diet and ocular diseases, as in the present work. In fact, our review article included four subsections regarding the benefits of natural food for vision health. Broccoli, nuts, saffron and tigernuts are awesome single foods that can help prevent/manage ocular diseases and also can help fight against certain risk factors related to visual impairment. Practically all of these food display anti-inflammatory, detoxicating, anti-angiogenic, anti-apoptotic, photo-protective, antioxidant and neuroprotective effects to some extent. Because of this, here we sought to address the role of diet and nutrition in the eyes and vision, focusing on the potential benefit of natural food (broccoli, saffron, tiger nuts and walnuts), as well as the benefits of the MedDiet and the positive effects of appropriated nutraceutical supplements on eye health in order to prevent vision loss.

## 3. Material and Methods

### Design

An extensive systematic search of PubMed, Web of Science, Scopus, Google Scholar, hand-searched publications and historical archives were performed by the professionals involved in this review, according to the standardized search of publications with the following keywords: eye, health, disease, vision loss, dry eyes, glaucoma, cataracts, retinopathies, macular degeneration, blindness, natural food, risk factors, pathogenic mechanisms, antioxidant, anti-inflammatory, neuroprotective, broccoli, saffron, tigernuts-*chufa de Valencia*, walnuts, Mediterranean diet (MedDiet), and nutritional supplements. Moreover, we utilized several combinations of the above terms. The main objective was to include English language peer-reviewed reports in which natural food, nutrient content, and its potential relationship with ocular health and disease were treated. Additionally, other important documents were revised in its native language. Five ophthalmologists and two researchers collected the characteristics, quality and suitability of the above studies, to ensure as much as possible the scientific interest and quality as well as to minimize the risk of bias. The MeaSurement Tool to Assess systematic Reviews (AMSTAR-2), an important tool for critically appraising systematic reviews of randomized controlled clinical trials) [59], the risk-of-bias (ROB) assessment of systematic reviews [60] and the Appraisal of Guidelines for Research and Evaluation (AGREE II), the most widely used guidelines appraisal tool [61] were utilized. Finally, 177 papers, from a period between 1983 to 2021 were selected for this study, mainly related to natural food, MedDiet, nutraceutic supplementation and methodology in the context of ocular health and the most prevalent eye diseases. Original studies from our group done with broccoli, and tigernut (*chufa de Valencia*) regarding the ocular surface dysfunction, macular degeneration, diabetic retinopathy and glaucoma were enclosed.

The specific methodology that was followed on the not previously published studies dealing with the effects of daily intake of broccoli and tigernuts, were precisely described in the corresponding subsections.

## 4. Natural Food and Ocular Health

### 4.1. Broccoli

The *Brassica Oleracea*, belonging to the vegetable *Brassicaceae* family, is popularly known as broccoli (Italian variety), brecol (spanish) and ka-i-lan or kale (Chinese variety) with the new food baby broccoli or tenderstem being a mix between the traditional broccoli and the kale. This family of natural foods constitutes a group of vegetables including the following: cabbage, broccoli, cauliflower, red cabbage, Brussels sprouts, radish, turnip, and others, all of them important sources of micronutrients and fiber [62]. The green broccoli (Calabrese) is the most common variety of this plant, sized 10–20 cm, weighing 500 g, with dark green and light green stems and buds from chlorophyll pigment. The broccoli contains high levels of water, carotenes (beta-carotene, lutein), vitamins (A, B, C, E), isothiocyanates, fatty acids (linoleic acid, palmitic acid) and diverse minerals (calcium, iron, magnesium, potassium, phosphorus, sodium). Additionally, amino acids (tyrosine, aspartic acid, glutamic acid, proline, valine) were found in larger concentrations. Even more, broccoli sprouts, have been implicated in many biological activities such as antiapoptotic, anti-inflammatory, antioxidant, antimicrobial, as well as neuroprotectant properties, as recently reviewed [63,64,65]. The ethyl acetate fraction of broccoli florets was reported to exert potent antioxidant and anti-inflammatory effects, by inhibiting nitric oxide release, counteracting the ROS and nuclear factor-κB activation in a dose-dependent manner, concluding that broccoli can be utilized as a dietary supplement to improve nutrition as well as for the adjunctive intervention in chronic inflammation [66]. The pro-apoptotic function of broccoli has been previously demonstrated in different cancers [67,68,69]. In this concern, the effects of the bio-accessible fraction of broccoli, kale, mustard, and radish, were evaluated on colon cancer cells, showing its usefulness to reduce this disease in combination with a balanced diet. However, a review about the targets and mechanism on breast cancer reported the contradictory roles of sulforaphane derivatives in breast cancer therapy [70]. The effects of the broccoli isothiocyanates, amino acid compounds that are detoxified by conjugation with glutathione, have also been reviewed in both in vitro and in vivo models of acute and chronic neurodegenerative diseases [71]. Similarly, the sulphoraphane (glucoraphanin), a phytocompound belonging to the isothiocyanate family with a role in preventing vascular complications in diabetes [72], also demonstrated important benefits for neurodegenerative disorders [73].

Nowadays, AMD in the dry and wet clinical types is the first cause of blindness among the elder population [2,5,6,42,56]. It has been estimated that early AMD cases will augment to approximately 17.8 million in 2050 [74]. Increasing the consumption of specific nutrients may be an effective intervention to vision care in AMD as well as in other sight-threatening ocular diseases. Among these nutrients, the xanthophyll pigments lutein and zeaxanthin and its metabolic by-products, were identified in the macula in 1985, at the highest concentrations of the whole human body, suggesting pivotal roles for these carotenoids in the retina [75]. In fact, lutein and zeaxantin have been proved to anatomically and functionally protecting the macula against photo-oxidative attack [76]. The Age-Related Eye Disease Study (AREDS) concluded that the daily intake of 10 mg of lutein and 2 mg of zeaxantin alone or in combination with docosahexaenoic acid (DHA) (350 mg/day) and eicosapentaenoic acid (EPA) (650 mg/day) to the original AREDS supplement formula composed by vitamin C, vitamin E, beta carotene, zinc oxide and cupric oxide [77], except the beta carotene, demonstrated efficacy in the prevention of AMD progression to advanced forms in right risk eyes [78]. The question arose as to whether it is possible to identify individuals at risk of AMD based on the findings of their central macular pigment optical density (MPOD) levels. In this concern, Berstein et al. [79] concluded that a central MPOD below 0.2 d.u. should be taken as low levels, 0.2–0.5. d.u. as mild levels and more than 0.5 as high levels.

To improve knowledge of the effects of the intake of broccoli on ocular health our group performed a pilot intervention study involving 14 voluntaries age/sex-matched, divided into: (1) healthy participants consuming a daily amount of 375 g of broccoli, which is equivalent to 10 g of lutein [77], according to a strict menu plan lasting for 4 consecutive weeks, (BG; *n* = 7), and (2) healthy participants no broccoli consumers, as the control group (CG; *n* = 7). We determined plasma total antioxidant capacity (TAC), by enzymatic-colorimetric assays, as previously published [6,7,8,9,10,11] and MPOD determined from retinographies from the right eye (RE) and left eye (LE) collected with the VISUCAM 500^®^ (Carl Zeiss Meditec Iberia, Tres Cantos, Madrid, Spain). This tool was gently rendered by Carl Zeiss Meditec Iberia (Tres Cantos, Madrid, Spain). As previously, reported [80]. An important requirement was to respect the way of cooking to avoid loss of the broccoli properties, including not eating the broccoli boiled, fried, or steamed and not using oven and microwave, according to the Yu et al., reports on the impact of home food preparation on the availability of antioxidants and other bioactivities in broccoli [81]. Volunteers were instructed not to smoke and not to abuse alcohol. We also advised the participants to avoid eating citrus (lemon, tangerine, orange, kiwi), carrots, spinach, or beans. Data showed an improvement of the subjective criteria on visual function in the activities of daily living. A significant increment in the MPOD in the retinographies of the RE from the BG (Figure 1) by an average of 30% more than their counterparts not assigned to the broccoli course.

Moreover, it was also found a significant increase in plasma TAC in the BG (baseline: 1.231 ± 0.120 mM; end-of-study: 1.858 ± 0.393 mM; *p* = 0.002). This study mainly suggests that the broccoli intake improved the antioxidant load and the MPOD associated with lutein dietary supplementation, thus helping in protect the macula against oxidative injury.

In summary, the biochemical and physicochemical characteristics of broccoli make this food optimal to fight against age-related chronic inflammatory and/or neurodegenerative disorders, to better eye and vision care, as widely suggested [62,63,64,65,66,72,73,80,81,82,83,84,85].

### 4.2. Saffron

The *Crocus Sativus* is a plant that provides a spice, saffron, which has been classically used in food preparation, being the most expensive spice in the world [86]. Etymologically saffron comes from the Arabic term za’farān (yellow) as well as from the Persian za’ferân (golden stigmas), the Latin word safranum and the Spanish azafrán. Since ancient times, medicinal properties have been attributed to this species because it has more than 100 metabolites in the composition of its stigmas, including crocin isomers, zeaxanthin, lycopene and vitamin B12, among others [87].

Major saffron components are crocin and crocetin (that gives the yellow color to the stigmas), picrocrocin (that contributes to the bittersweet flavor), kaempferol (from the crocus sativus petals) and safranal (which lends the fragrance to the spice, also contributing to the flavor) [86,87,88]. However, the main therapeutic activities of saffron are due to its main bioactive components, the carotenoids crocetin and crocin [88]. Crocin is hydrolyzed to crocetin when absorption occurs in the intestine [89], and once in the blood it can be transported to different tissues and can even cross the blood-brain barrier reaching tissues of the central nervous system [90]. This spice has been used in traditional medicine as anti-ischemic, hypolipidemic, anti-hypertensive, anxiolytic, antidiabetic, antidepressant, anticancer, and cardioprotective [91,92]. It has been possible due to the various properties attributed to crocetin as an anti-inflammatory, anti-apoptotic and antioxidant [92]. Antioxidant activities are due to their ability to scavenge free radicals [93], their capacity to decrease telomerase activity and to increase proapoptotic effects in cancer cells. In addition, the anti-inflammatory effect is due to the regulation of genes that control the release of proinflammatory cytokines, adhesion molecules and proinflammatory enzymes by glial cells, as well as modulation of inflammatory pathways (e.g., nuclear factor-κB) [94].

The beneficial effects of saffron have been demonstrated in neurodegenerative diseases such as Alzheimer’s and Parkinson’s, where it has been shown to exert a certain neuroprotective effect [95,96]. In addition, in neurodegenerative diseases of the eye, saffron may also have these beneficial effects [94].

In AMD, it has been shown that in the early stages of the disease, saffron can improve visual function by reversing the damage to photoreceptors and bipolar cells caused by oxidative stress [97]. In addition, daily supplementation with saffron improved retinal changes observed with optical coherence tomography and electroretinogram in patients with both dry and wet AMD [98].

In DR, saffron can reduce insulin resistance in pre-diabetic patients [99]. In vitro models of diabetes have shown that saffron can control microglia activation. In addition, crocin supplementation decreases macular thickness improving visual acuity in patients with diabetic macular edema probably by its anti-inflammatory effects [100].

In glaucoma, there are two studies in patients on the possible hypotensive effect of saffron that show contradictory results. In patients with primary open-angle glaucoma, using a dose of 1g twice a week, no hypotensive effect was observed [101]. However, in another study using a dose of 30 mg per day, a decrease in intraocular pressure (IOP) was found [102]. Our group recently analyzed the anti-inflammatory and neuroprotective effect of saffron in a mouse model of chronic ocular hypertension (OHT) [103]. We found that saffron extract (standardized to 3% crocin content) resulted in a reduction in both the number and signs of microglial cell activation (Figure 2A,B) as well as a down-regulation of the purinergic receptor P2RY12a, a marker of inflammation-related non-activated microglia. In addition, saffron also prevented the retinal ganglion cells (RGCs) death that occurred in chronic hypertensive eyes (Figure 2C,D), postulating that this neuroprotective effect of saffron could be due to its anti-inflammatory and antioxidant properties [103].

### 4.3. Tigernut-Chufa de Valencia

The *Cyperus Esculentus* is a herbaceous, perennial, fasciculate-rooted plant found across the world, but distributed mostly in Egypt, Nigeria and Spain. Additionally known as “Juncia Avellanada” and tigernut (TN), it has a highly developed rhizomatic system with the tubers produced at the apical ends of the rhizomes [104]. It is known that only one specimen can produce hundreds or thousands of tubers through a growing season. According to the macroscopic characteristics three main tuber types have been described: brown, black and yellow [105]. Over the last decades, substantial research has evidenced that TN is a good source of oil, and its by-products are rich in various nutrients and bioactive compounds [105,106]. The available data reveals that tubers are rich in essential dietary constituents such as proteins (3.28–8.45%), fats (22.14–44.92%), fibers (8.26–15.47%) and ashes (1.60–2.60%). The lipid profiling interestingly revealed that TN oil has a similar fatty acid composition to olive oil [107].

The TN finds the Mediterranean climate of Valencia particularly favorable for its cultivation and development. In this area, it is known as *chufa de Valencia*. It was introduced in the Valencian region in the 8th century CE, a multi-cultural period lasting 711–1492 in which Christians, Jews and Muslims created a high degree of civilization in Spain and Europe. The curative properties of the *chufa de Valencia* date from 1297 [108]. Arnau de Vilanova (1232–1311), a famous physician and theologian of this time, prescribed eating *chufa de Valencia* to alleviate different disorders [109]. The Valencian botanist Cavanilles (1745–1804) reflected in his works from 1795, the cultivation of the TN in the town of Alboraya (Valencia, Spain) [110]. The *chufa de Valencia* is dark brown, sized 0.9-1.6 cm long and 0.7–1.1 cm wide. According to the shape, two types are distinguished: the elongated (*chufa llargueta*) and the rounded (*chufa armela*). The *chufa de Valencia* is a historical gastronomic and cultural wholesome brown tuber crop, with outstanding nutritional properties. This fresh functional food consists in carbohydrates 18%, fats 17% (including ω6/ω9 fatty acids), proteins 8%, fiber 13%, oligoelements (calcium, copper, iron, magnesium, phosphorus, potassium, zinc) and vitamins (C, E), providing 460 kcal/100 g [111]. It is also extensively used to prepare a cold beverage, known as “horchata de chufa” typical to Valencia [112].

Several epidemiologic and experimental studies pointed out the wide variety of therapeutic benefits of TN, such as cardioprotector, antioxidant, anti-inflammatory and neuroprotectant [113,114]. The TN also contributes to lowering total cholesterol and triglycerides, stabilizes glycemic profile, provides amino acids, vitamins, minerals, and fiber. Its salt content is low, and it does not contain lactose or fructose. Thus, the TN and its derivatives (*horchata*, flours, oils, spices, etc.) constitute a very complete food, as they offer large proportions of vitamins and minerals (such as vitamins C and E), lipids and oleic acid which are useful for the control of cholesterol and triglycerides [115,116]. As for the presence of vitamin E, it is important to highlight its importance as it is an essential vitamin, not synthesized by the body, but necessary for its proper functioning, and therefore must be included in the diet [117]. In addition to being one of the major antioxidants, vitamin E has the ability to scavenge free radicals, which reduces the risk of cancer and prevents the progression of pre-cancerous lesions [118,119].

Among the most prevalent eye diseases are ocular surface disorders. DED defines the pathology of the ocular surface that induces tear-film deficiency and dry eyes [120]. The term includes complex diseases that affect the eyelids, lacrimal glands, conjunctiva, cornea and the tear film, with very high global prevalence, affecting both genders and people aged 60 years and above. The signs and symptoms range from mild redness, foreign body sensation, photophobia, and/or blurred vision, to intense and diffuse hyperemia, epiphora, continuous sensations of itchiness, stinging and burning, as well as visual impairment. First-line therapy includes eye drops of artificial tears, lubricant gels and ointments. The imbalance between prooxidant and antioxidant sources damages the ocular surface structures [121,122]. It has also been assumed that chronic inflammation is involved in OSD/DED, as demonstrated by the release of pro-inflammatory mediators and the positive response to the oral supplementation with antioxidants and essential fatty acids [123,124]. In this concern, anti-inflammatory eye drops can also be prescribed [125,126]. In spite of this, further research is needed to improve the eyes and the quality of life of patients affected by DED.

In this concern, our group conducted one study in the past years (2016–2018) about the role of the daily intake of *chufa de Valencia* in eye health, with the main purpose of evaluating its effects on DED by integrating clinical and biochemical data. A pilot study on 20 women aged 45–70 years, office employees of the administration services of the University of Valencia, with the common characteristic of working with computers during the workday were included in the study and classified as: (1) women working with computers assigned to a daily ration of 30 g of fresh *chufa de Valencia*, kindly given by the Regulatory Council of the Designation of Origin *Chufa de Valencia* (Alboraya, Valencia, Spain) during 3 consecutive months (*n* = 10; ChG) and (2) women working with computers without consuming the tuber (*n* = 10; CG). A personal interview including the ocular surface disorder questionnaire (OSDI; Allergan Inc., Irvine, CA, USA) to discriminate between normal-mild-moderate-severe DED, and ocular examination (best corrected visual acuity, the spontaneous number of closing eyelids in 1 min: blinking frequency that in normal conditions 9–12/min; quantitative Schirmer test, to evaluate the amount of wetting the strip located on the inferior inner eyelid during 5 min, that in normal conditions is more than 10 mm; qualitative break up time test (BUT), the time interval between last blink and the appearance of first dry spot over the cornea, that in normal conditions is more than 5 s), were carried out for all women participants. One important point of this study was to ensure compliance with the supplement food by the participants, which is essential to optimize the effectiveness of the nutritional intervention. Average duration of computer uses during the workday was 5.8 ± 2 h with similar type of screen and computer for the two groups of participants. Tear samples from the inferior eyelid lacrimal meniscus were collected with capillaire microtubes, labeled and stored at −80 °C until processing.

Mean age of participants was 55.4 ± 6.2 years. The OSDI questionnaire revealed that 68.4% of the ChG had moderate dry eyes at baseline and the same participants had mild-to-moderate dry eyes at the end of study (61.6%). All volunteers displayed signs and symptoms of DED, ranging from redness, grittiness, itchiness, foreign body sensation, burning, stinging and blurred vision. Eye discomfort, visual impairment, and reduction of the quality of life related to the eyes and vision, were referred by the volunteers at the onset of this study. However, a noticeable reduction of the signs, symptoms and subjective sensations was recorded at the end of the food supplementation. The blinking frequency was significantly and positively reduced in the ChG after the oral intake period as compared to the non-supplemented employers (*p* = 0.042). The BUT test was significantly higher in the ChG at the end of study (RE: 7.4 ± 0.7 s, vs. 9.8 ± 0.4 s; *p* = 0.011; LE: 7.5 ± 0.7 s vs. 9.7 ± 0.4 s, *p* = 0.016). Additionally, the ChG showed higher Schirmer test marks at end-of-study, as compared to baseline (RE: 7.1 ± 0.7 mm vs. 10.5 ± 0.9 mm, *p* = 0.002; LE: 7.0 ± 0.6 mm vs. 12.9 ± 1.8 mm, *p* = 0.001), as reflected in Figure 3. In addition, no adverse effects were reported in relation to the supplementation in the assigned participants to this regime. In contrast, all participants declared to be satisfied with the organoleptic properties of the intake of *chufa de Valencia*.

In conclusion, the daily intake of 30 g. of *chufa de Valencia* improved the amount and stability of the tear film, decreasing the signs, symptoms, and subjective sensations of the DED patients.

As reflected in the anterior subsection (3.1 Broccoli) of this review, AMD refers to the chronic, progressive degeneration of the macula, a common eye disorder affecting people aged 60 years and more [1,2,3,6,36,37,38,77,78,79]. Up to 200 million people worldwide currently have AMD which is caused by complex interactions between aging comorbidities and genetics with the environmental factors, and other unknown situations [1,2,3,6]. The clinical AMD forms, dry and wet, can be clinically distinguished, with the dry AMD accounting for 90% of all cases. AMD progressively leads to central vision loss and reduced quality of life in the affected individuals.

We performed onr more pilot study on 30 healthy volunteers (12 men and 18 women) aged 44–51 years to evaluate the effects of the daily intake of fresh *chufa de Valencia* on the antioxidant status as well as in the MDOP measurement. As in the previous work, the tubers were gently donated by the Regulatory Council of the Designation of Origin Chufa de Valencia (Alboraya, Valencia, Spain). Each participant consumed a daily ration of 30 g of fresh *chufa de Valencia* during 3 consecutive months. It was not permitted to eat spinach, broccoli, pumpkin, carrots, citric, horchata or oral/topical nutraceutics, during the study course. The MPOD was done at baseline and at end of follow-up. Retinographies from the RE/LE were collected with the VISUCAM 500^®^ (Carl Zeiss Meditec Iberia, Tres Cantos, Madrid, Spain). Blood samples were collected from the antecubital vein, that were centrifuged to separate blood and plasma. The latter was aliquoted and frozen at −80 °C until processing to determinate the malondialdehyde (MDA)/thiobarbituric acid reactive substances (TBARS) and TAC, by enzymatic-colorimetric methods.

The *chufa de Valencia* daily core regimen induced a significant increase in the MPOD as reflected in the retinographies from baseline to end-of-study (Figure 4).

Additionally, a significant improvement of the plasmatic TAC values (*p* = 0.032) and a significant reduction in parallel of the plasmatic MDA/TBARS (*p* = 0.017) at the end of follow-up was clearly detected (Figure 5). From this latter study, we concluded that the daily intake of fresh *chufa de Valencia* mitigated the oxidative stress by means of its antioxidant effects, as well as protected the macula by increasing the MPOD in healthy subjects.

These findings suggest that a regular *chufa de Valencia* intake may serve as a dietary prophylaxis adjunctive intervention for patients at risk of AMD and vision loss.

### 4.4. Walnuts

Walnut is the fruit of the juglans tree (*Juglandaceae family*), with strong, outspread branches, native probably from Persia (*Juglans Regia*). The walnuts are the round seeds of that tree, available in a range of distinct sizes and colors, and usually consumed as a nut when the hard shell has been wide open and discarded. Walnuts are a rich source of bioactive nutritional components that has the ability to modulate multiple metabolic pathways that contribute to protection against many chronic diseases, including those that affect the optic nerve, the retina and the ocular microcirculation. Most of these bioactives have a synergistic effect, acting in the protection of the physiological metabolic and vascular pathways [125]. Walnuts have high contents of polyphenols, phytosterols, Υ-tocopherol, and mainly α-linolenic acid (ALA), in addition to minerals [125], as listed in Table 1 for the *Juglans Californica*, being the general properties antioxidant, anti-inflammatory, neuroprotection, anti-thrombotic, anti-arrhythmic, cardiovascular protection, cholesterol-lowering and improve gut microbiota.

Human clinical trials have suggested an association of walnut consumption with better cognitive performance and memory improvement in adults, with beneficial effects on memory, learning, motor coordination, anxiety, and locomotor activity [126,127,128]. These studies also concluded on the benefits of a walnut-enriched diet in brain disorders and other chronic diseases [126,127,128,129]. The additive effect of the essential components of walnuts is proven, with protective action against the events related to oxidative stress and inflammation present in different chronic diseases [125,130]. All these positive health effects can be obtained in different eye diseases, such as glaucoma, DR and age-AMD [131], chronic pathologies with a degenerative character for the ocular structures, which share common pathophysiological mechanisms, characterized by the presence of events related to oxidative stress and inflammation. Likewise, in recent decades, with the increase in life expectancy and the progressive growth of the population with its consequent aging, we have noticed a significant increase in the incidence of chronic neurodegenerative disorders, as in the case of AMD. Its socio-economic consequences are evident, both in terms of the decrease in the quality of life of those affected and in terms of a considerable pressing in the health care system and increased financial burden.

Recent experimental evidence suggests that the main *polyphenols* of walnuts, ellagitannins and their metabolites (*urolithins*), have beneficial properties against the oxidation processes of cellular components and in the inflammation pathways, in addition to positively influencing the intestinal microbiome [125,132]. *Phytosterols* have proven antioxidant properties and are partly responsible for their cholesterol-lowering effect. They are powerful free radical scavengers, acting to reduce pro-inflammatory eicosanoids, and then mitigating the inflammatory response [130,131].

The metabolism of ALA—the vegetable ω3 fatty acid—gives rise to vasodilator and anti-inflammatory *oxylipins*, which can be the basis for a protective action on the function of capillary endothelial cells. Its neuroprotective capacity has also been described on brain function, inducing vasodilation of the cerebral arteries with improved irrigation and contributing to phenomena related to neuroplasticity. These effects could also be observed in the retina and the optic nerve [133,134,135,136,137,138]. Interestingly, in addition to its already known anti-arrhythmic potential, ALA can exert other beneficial effects on cardiovascular function, through an anti-thrombotic, anti-inflammatory, and cholesterol-lowering action. The latter are considered protective factors against atherosclerosis [137]. On the ocular tissues, its vasculoprotective action could contribute to an improved endothelial function in the microcirculation of the retina, the cribriform plate and the choriocapillaries [137,138].

Finally, walnuts are also rich in Υ-tocopherol—a form of vitamin E—as we have explained before, a powerful antioxidant with anti-inflammatory properties, with protective and preventive action in macular diseases, such as AMD [139]. Non-sodium minerals such as potassium, calcium and magnesium, shared by all nuts, and especially in walnuts, also have a protective effect on cardio-metabolic risk, as confirmed by recent evidence [138,139].

Primary prevention in many of these neurodegenerative diseases is crucial from the point of view of public health and could be achieved early in life by introducing a healthy diet, rich in antioxidant and anti-inflammatory phytochemicals, as is the case with dietary supplementation with walnuts, as for its nutritional value for ocular chronic diseases.

## 5. Mediterranean Diet—Current Knowledge on Eye Diseases

Dr. Keys, a physiologist from Minnesota (USA), analyzed in the second half of the twentieth century the nutritional habits and lifestyle of people settled in the European Mediterranean countries. He conducted various larger population studies to find a noticeable reduction in the incidence of cardiovascular and other chronic disorders, altogether with a higher lifespan of this area. In his renowned publication of “the seven countries study,” [140] he hypothesized basically that cholesterol fat and the intake of food containing cholesterol are important hallmarks of morbimortality. Among the milestones of the MedDiet are the following: (1) the elevated consumption of bread, cereals, olive oil, fruits, legumes, and vegetables altogether with the low consumption of saturated fat; (2) the moderate-to-high intake of chicken and fish; (3) the moderate intake of cheese, yogurt, and wine; and (4) the low consumption of red meat. In addition, a healthy lifestyle should be followed, doing physical activity regularly. Its beneficial effects on health motivated its appointment of Intangible Cultural Heritage of Humanity in November 2010. Therefore, the United Nations Educational, Scientific and Cultural Organization (UNESCO) raised the MedDiet as a lifestyle and cultural heritage for humanity.

Numerous studies have been conducted on its protective effects against chronic cardiovascular-inflammatory-metabolic- neurodegenerative diseases, such as cardiovascular disorders [140,141,142], diabetes mellitus [143,144], obesity [145,146], cognitive decline/dementia [147,148], and Alzheimer disease [149]. Additionally, there are studies showing the possible protective effect of MedDiet in age-related ocular pathologies.

In this sense, Raimundo et al. [150] carried out a nested case-control study within the Coimbra Eye Study in patients with AMD. They used a food consumption frequency questionnaire and a scale initially developed for the Greek population (mediSCORE). The authors concluded that high adherence to a MedDiet protects against AMD due, mainly, to the high intake of antioxidants. Keenan and colleagues [145] found similar results in a retrospective cohort-based study of the AREDS and AREDS2. In this case, the authors used the Alternative Mediterranean diet Index (aMedi) based on food frequency questionnaires and observed a lower risk of AMD progression in those patients with high adherence to MedDiet.

There is also evidence of the protective effect of the MedDiet on the onset of DR. Díaz-López et al. [151] conducted a nutritional intervention study in patients with T2DM who did not have microvascular complications at the beginning of the study. Three dietary interventions were analyzed: MedDiet supplemented with extra virgin olive oil, MedDiet supplemented with walnuts and a low-fat control diet. After a 6-year follow-up, it was found that MedDiet enriched with extra virgin olive oil had a protective effect on the development of DR. Other research carried out within the Prevention with the Mediterranean Diet (PREDIMED) study concludes that 500 mg/day of ω3 fatty acids, an amount easily achievable with good adherence to MedDiet, significantly reduces the risk of developing DR [57] and mortality was analyzed under dietary α-Linolenic Acid and marine ω3 fatty acids [152].

Regarding glaucoma, Abreu-Reyes et al. [153] conducted an interesting observational study in the Spanish Canary Islands on the adherence to MedDiet in 100 patients with primary open-angle glaucoma (POAG). The authors reported moderate adherence to MedDiet in 71% of the subjects. It would be advisable to carry out more intervention studies to estimate the potential benefit of MedDiet in the glaucoma risk and progression.

In the Valencia study of diabetic retinopathy (VSDR) the influence of MedDiet in nutritional outcomes of type 2 diabetics with or without DR were analyzed [40,154]. According to the PREDIMED study [57] a 14-item questionnaire to assess the adherence to the MedDiet of T2DM patients was applied, to be compared with those from the healthy controls (CG). Scores indicating compliance to MedDiet distinguished between those participants with a high intake of bread, cereals, fish, fruits, legumes, olive oil, vegetables, and red wine that were positives (1), while those with a lower intake of the above foods were negatives (0). Data from this questionnaire showed average values significantly higher in the CG (9.8 ± 2.1) versus the T2DM patients (6.4 ± 1.1) (*p* < 0.05). The comparison of the prooxidant and antioxidant markers analyzed in the present study in the T2DMG with poor and good adherence to the MedDiet is reflected in Table 2.

Controversial results have been reported on the effects of the MedDiet regarding DED, cataracts, and glaucoma. Molina-Leiva and the PREDIMED PLUS study [155] concluded that implementing the MedDiet and lifestyle pattern to benefit dry eye patients.

A prospective case-control study was done by our group in 2016–2017, in collaboration with the Department of Ophthalmology of the Careggi Hospital (University of Florence, Italy), including 112 consecutive individuals of both sexes aged 25–80 years, to address the adherence to the MedDiet in an ophthalmologic population of Valencia (Spain) [156]. Patients were included on the basis of the diagnosis of DR, AMD, POAG, or cataracts. By means of the slit lamp (ImageNet, Topcon, Barcelona, Spain) the ocular fundus examination was done to confirm DR diagnosis, and biomicrosopy of the anterior eye segment and media was done to confirm cataracts diagnosis. The POAG group was addressed by the IOP elevation (>21 mm Hg) according to the central corneal thickness values, as well as by morphologic/structural and functional probes (optic disc examination, optic coherence tomography (OCT) and visual field performance), as suggested by Castejon-Cerveró et al., over the European Glaucoma Society Guidelines for Spain [157]. The AMD was diagnosed on the basis of the ocular fundus and OCT examination as suggested by the Spanish Society of Retina and Vitreous (SERV) guidelines [158]. Personal interviews and the 14-items validated MedDiet questionnaire were done. Additionally, socio-demographic data, risk factors for ocular diseases and ocular examination were performed. The questionnaire scores indicating compliance to the MedDiet is able to distinguish between participants with a high intake of cereals, legumes, fruits, vegetables, olive oil, fish, nuts, industrial pastry, wine and meat (red or white meat, cold meat) and those with a low intake of these foods. Good adherence to the MedDiet is scored with a total of 9 points of 14. Mean age of the suitable participants was 55.8 ± 14.1 years (58,7 % females and 41.3% males). Average score of the questionnaire was 8.4 ± 2.2 suggesting suboptimal adherence for almost half of the selected participants without differences across the cases and the controls (Table 3).

No statistical differences were found in adherence to Mediterranean diet by age (*p* = 0.536) or sex (*p* = 0.104) in the participants suffering the above ocular diseases. Moreover, no individual item differed significantly between the analyzed ocular disease patients except for a trend towards a larger consumption of fish.

This work concluded that adherence to MedDiet was suboptimal in half of patients, regardless of their ocular disease status, including diagnoses of cataract, glaucoma and DR. Data suggested that educational interventions to improve the dietary habits of ophthalmic outpatients could be investigated in further prospective studies.

In summary, MedDiet may be considered a protective factor against the onset of the leading causes of blindness. Further investigations are essential to deep on the knowledge of benefits of this diet to appropriately recommend it to the population, and to assess the possibility of using nutritional supplementation in cases of low-medium adherence to a MedDiet, to uncover methods for prevention of vision loss and quality of life.

## 6. Nutritional Supplements

The most prevalent sight-threatening ophthalmic diseases are DR and glaucoma, leading causes of vision loss in the global population. Dietary habits can contribute to DR initiation and progression. Lower dietary fiber is associated with increased risk of developing DR [159]. Plus, once these complications appear, intensive glycemic control has been demonstrated to reduce the rate of diabetic DR [158]. MedDiet, characterized by being reach in fruits and vegetables and unrefined carbohydrates, has been related to reduced risk of DR incidence [151,160].

Concerning glaucoma, two epidemiological studies included in the Study of Osteoporotic Fractures found a decrease in risk of developing glaucoma when high consumption rates of green leaves and vegetables, fruits and fruit juice were consumed [161,162]. Recent systematic reviews found that selenium and iron may increase the risk of glaucoma but the relationships of dietary intake of other substances did not present a strong association with the risk of glaucoma [50] and a beneficial association among dietary intake of vitamin A and C with open angle glaucoma [163].

There is an increased interest in nutritional supplementation as coadjuvant therapy in DR and glaucoma patients. Different types of methodologies and supplements, mainly vitamins, have been studied. Outcomes have been considered in terms of clinical parameters (visual acuity, IOP, global indices of visual field examination, degree of retinopathy or thicknesses at different ocular locations assessed by OCT. The most relevant investigations in this sense are mentioned below.

In relation to DR, Bursell et al. [164] investigated the supplementation with high-dose of α-tocopherol in a group of type-1 diabetes mellitus patients and found an increase of retinal blood flow that improved after 8-month supplementation. Combination of vitamins and other microelements have also been considered. A study by our group with five-year follow-up dealt with the effects of a complex composed of lutein, α-tocopherol, niacin, beta-carotene, zinc and selenium in 105 type-2 diabetic patients suffering from non-proliferative diabetic retinopathy (NPDR). We found a decrease in the clinical funduscopic progression but visual acuity did not change [165]. In contrast, Hu et al. [166] found an improvement of visual acuity, and also of contrast sensitivity and foveal thickness in type-1 and type-2 diabetic patients with NPDR after a 3-month supplementation with lutein ad zeaxanthin. Similarly, Chous et al. [167] demonstrated an increase of visual acuity with no alteration of retinal thickness with the administration during 6 months to in type-1 and type-2 diabetic patients with no retinopathy or NPDR of DiVFuSS complex containing vitamins C, D3 and E, zinc oxide, eicosapentaenoic acid, docosahexaenoic acid, α-lipoic acid (racemic mixture), coenzyme Q10, mixed tocotrienols/tocopherols, zeaxanthin, lutein, benfotiamine, N-acetyl cysteine, grape seed extract, resveratrol, turmeric root extract, green tea leaf, and pycnogenol. The VSDR group performed a prospective case-control study in 575 participants during 38-month follow-up on the effects of the daily intake of a pill containing antioxidants, trace metals and ω3 fatty acids (Nutrof Omega ^®^ formula), in T2DM patients, with and without DR and healthy controls, concluding that this course changes reduced the oxidative load in patients at risk of DR [40,153].

In relation to glaucoma, supplementation with black currant anthocyanins extract for two years was related to a better visual field performance in comparison with non-supplemented group in glaucoma [168]. Plus, this extract was proved to be related to a decrease of IOP in healthy and glaucoma patients [169]. In addition, supplementation with ginkgo biloba extract during a mean period of over 12 years showed a slower progression in perimetric global indices in normotensive glaucoma [170]. Alternatively, nutritional supplementation by our group with two complexes of vitamins and other substances (one with and one without ω3 fatty acid supplementation) on IOP-controlled primary open-angle glaucoma patients did not overcome control group in terms of visual field, peripapillary nerve fiber layer thickness or macular ganglion cell complex thickness in a two-year follow-up. Thus, we concluded that this kind of supplementation did not seem to be useful [171]. In contrast, Mutolo et al. [172] administered another complex (containing different vitamins such as B1, B2, and B6) and they found a decrease of IOP and an amelioration in eletroretinogram results in a 12-month follow-up in open-angle glaucoma patients. More recently, Romeo Villadoniga et al. [173] found that supplementation to pseudoexfoliative glaucoma patients with a complex containing DHA (and other components such as EPA, vitamins B, C, E, lutein, zeaxanthin, and minerals) was related to a decrease of IOP at 3 and 6 months. Moreover, Galbis-Estrada et al. [45] showed that POAG patients showed a lower tear expression of inflammation biomarkers when supplemented with antioxidants and ω3 fatty acids (Brudysec ^®^ formula) over 3 months.

Regarding the OSD/DED, Downie et al. [174] performed a Cochrane systematic review including 4214 participants from clinical trial son the role of the administration of ω3 and ω6 PUFAs in OSD/dry eyes, but they concluded that the results were uncertain and inconsistent. Additionally, Liu and Ji [175] performed a meta-analyses of the effects of oral administration of supplements containing ω3 and ω6 PUFAs for OSD/DED showing that significant improvements in clinical tests and individual-reported symptoms were found. Pinazo-Durán et al. [44] performed a 3-month follow-up on the role of the oral administration of antioxidants and ω3 PUFAs (Brudysec^®^ formula), in OSD/DED finding that the clinical and subjective sensations on dry eyes significantly ameliorated in the affected patients, with improvement of dry eye-related quality of life by ameliorating clinical tests and tear expression of inflammatory mediators. Galbis-Estrada et al. [176] analyzed the metabolomic signature of tears from patients with OSD/DED by 1H nuclear magnetic resonance spectrometry. These authors also found and an improvement of subjective dry-eye symptoms and clinical signs in relation to statistically significant changes in the metabolomic profile induced by the oral supplementation of a combination of antioxidants and ω3 PUFAs (Brudysec ^®^ formula), for 3 months [123]. Additionally, Pinazo-Durán et al. demonstrated that an eyelid creme containing DHA and taking advantage of its anti-inflammatory properties ameliorated the cytokine expression in tears and the subjective sensations of ocular surface relief in healthy contact lens users [177].

Overall, these studies showed that interventions related to diet and nutritional supplementations in DED, DR and glaucoma could be promising coadjuvant approaches to conventional therapies.

These mentioned investigations are summarized in the Table 4.

## 7. Concluding Remarks

Knowledge on nutrition interventions for the prevention of chronic eye diseases is far from complete, as reflected in this review article. In fact, the extended lifespan and the increased prevalence of sight-threatening diseases have a huge socio-economic impact on healthcare systems, individuals, and their families worldwide.

Ophthalmologists, researchers, and policymakers, have to pay special attention to the diet and lifestyle patterns of the population, to identify people at high risk of developing eye diseases, such as OSD/DEDs, glaucoma, DR or AMD, or those individuals especially vulnerable to ocular diseases progression. It is urgently needed to strengthen all preventive measures to counteract the increment of these diseases in order to avoid visual impairment and blindness, and the loss of quality-of-life linked to vision.

This review has been designed to shed light on the above topics, to improve the understanding of the nutritional hallmarks of natural food, the benefits of the MedDiet and the most appropriate oral supplements with vitamins, carotenoids and PUFAs for better eye and vision care. In the present review work, we clearly demonstrate that natural food is essential for the eyes and vision and these discoveries are common to different countries where the revised studies were conducted, but also important differences have been detected across the revised studies. The controversy over these studies dealing with nutrients, foods, and dietary patterns, and other related findings, and its role in the eyes hasintensified through the years. Based on the conclusions of this review, we may propose that neglecting the potential of natural food, the benefits of the MedDiet and the positive effects of appropriated nutritional supplements may result in high socio-economic costs regarding ocular health.

## Figures and Tables

**Figure 1 foods-10-01231-f001:**
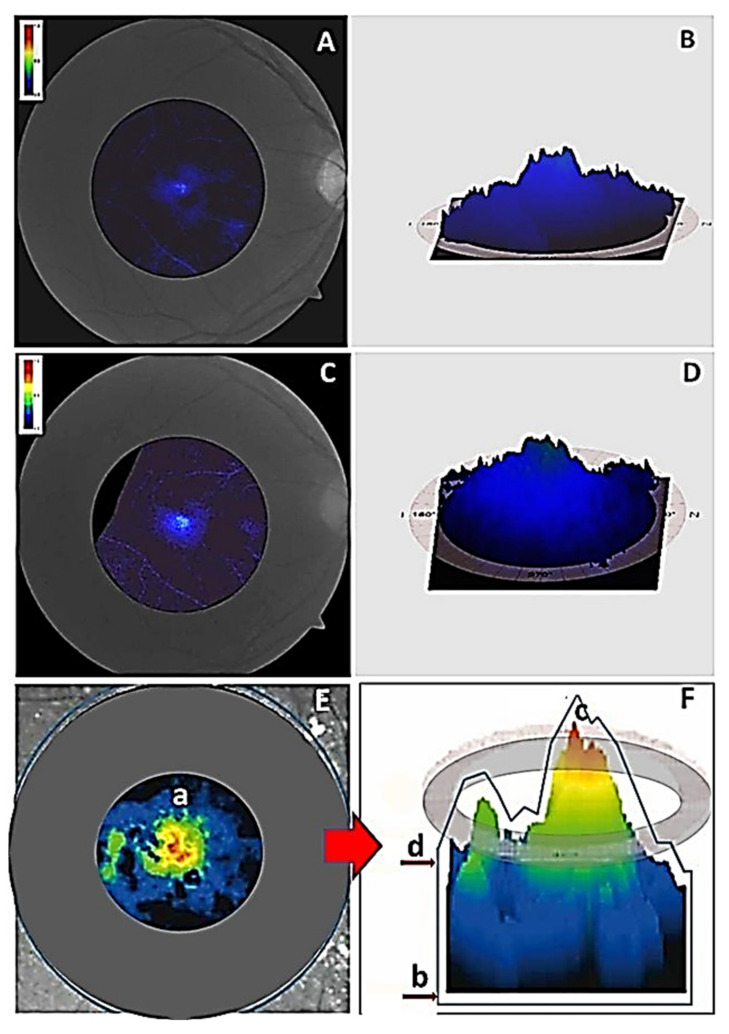
Comparative evaluation of the MPOD measured with the Visucam 500^®^. (**A**) Retinography of the central retina of the right eye at baseline, (**B**) Distribution of the macular pigment and the peak at the foveal level of the right eye at baseline. (**C**) Retinography of the central retina of the right eye of the same participant at end-of-study. (**D**) Distribution of the macular pigment and the peak at the foveal levels of the righ eye of the same participant at end-of-study (**E**) Schematic representation of the retinal area where the pigment is placed (a). (**F**) Illustrative drawing of the Visucam 500 parameters: total pigment volume (b), maximum of the pigment density (c), mean of the pigment density (d) [81].

**Figure 2 foods-10-01231-f002:**
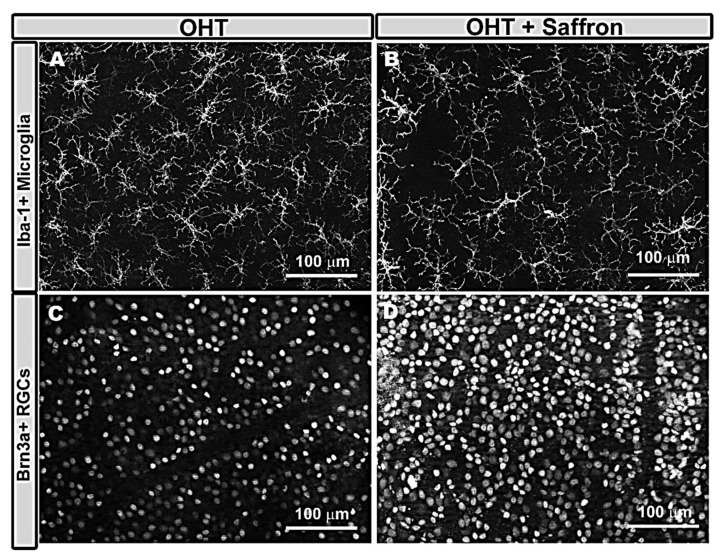
Immunohistochemistry of retinal whole-mount micrographs of mice eyes. The A,B retinas were labeled with anti-Iba-1 (microglia), showing the comparison of Iba-1 + microglia in the OHT untreated (**A**) and treated with saffron extract (**B**) eyes in the outer plexiform layer of the retina. It has been noted that in the OHT mice eyes treated with saffron extracts, less activation and fewer microglial cells were observed than in the untreated eyes. The C,D retinas were stained with anti Brn3a (RGCs). The C,D comparative micrographs showed the Brn3a + RGCs in the untreated (**C**) and treated with saffron extract (**D**) mice retinas. It was also detected a higher RGCs density in the treated versus the untreated rat retinas. RGC: retinal ganglion cells.

**Figure 3 foods-10-01231-f003:**
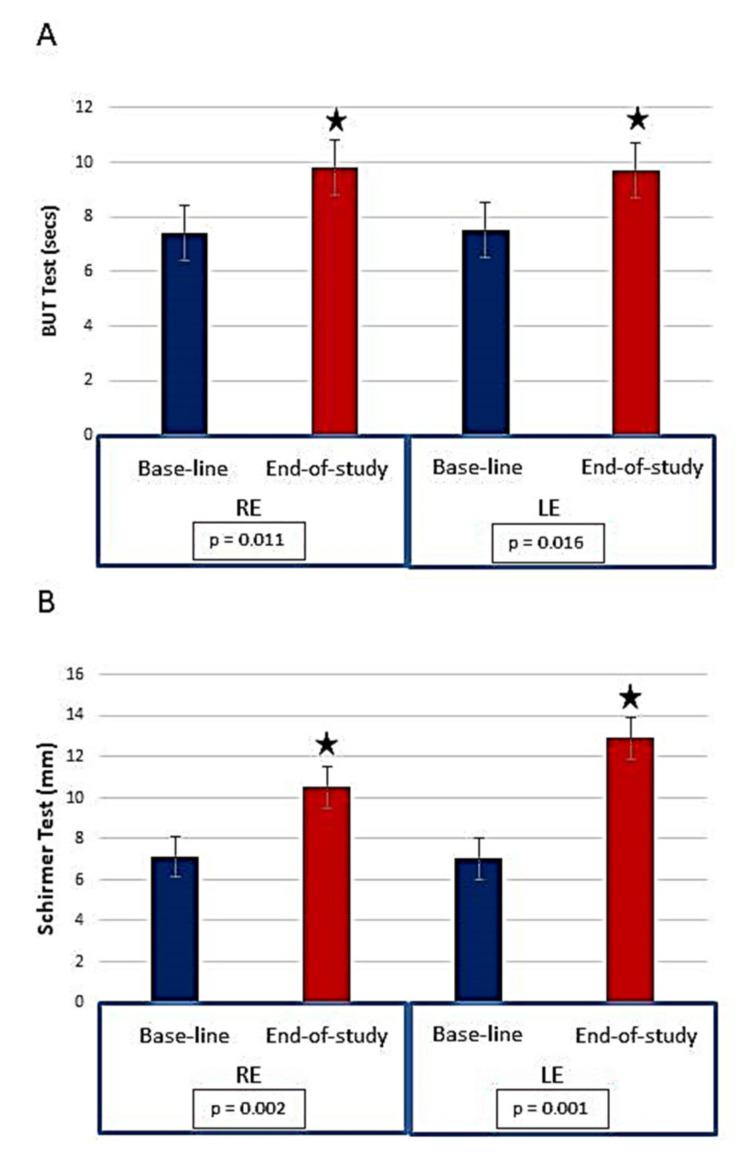
Clinical probes to qualitative and quantitative evaluating the tear film from baseline to the end of study in the participants randomly assigned to a daily intake of 30 g. of the fresh TN *Chufa de Valencia*. (**A**) Data from the FTBUT in both eyes. (**B**) Schirmer test determination in both eyes. F-BUT test: Fluorescein break up time test; RE: right eye LE: left eye. ^★^
*p* < 0.05 statistically significant.

**Figure 4 foods-10-01231-f004:**
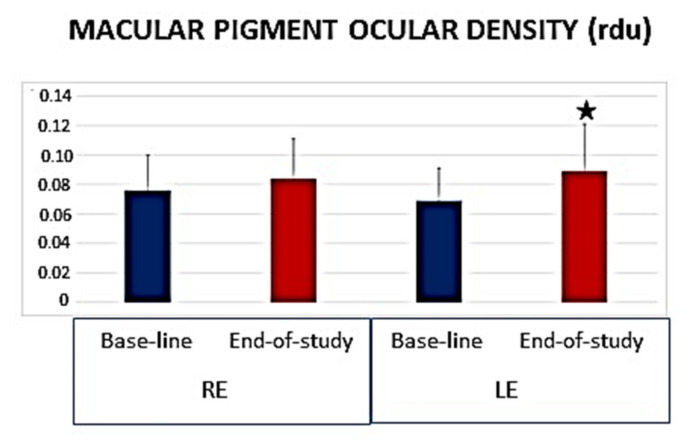
Parameters of the Macular Pigment Ocular Density. The volume is expressed in relative densitometry units (rdu). RE: right eye. LE: left eye. ^★^
*p* value < 0.05.

**Figure 5 foods-10-01231-f005:**
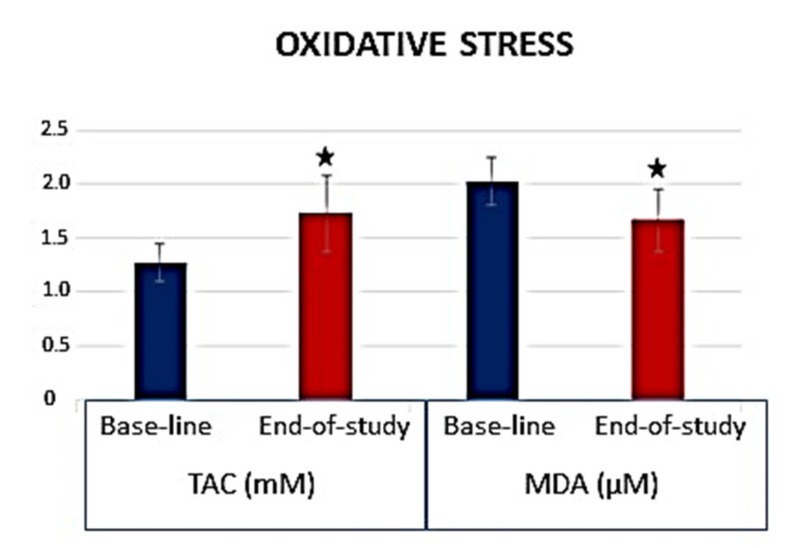
Parameters of Oxidative Stress in *Chufa de Valencia* intake, determined by TAC and MDA/TBARS. TAC: Total antioxidant capacity; MDA/TBARS: Malondialdehyde/thiobarbituric acid reactive substances; ^★^
*p* < 0.05.

**Table 1 foods-10-01231-t001:** Main components of walnuts and biomedical properties [125].

Walnuts (*Juglans Californica*)	
Content	Concentration/100 g
**Polyphenols**	2500 mg
**Phytosterols**	113 mg
**α-linolenic acid**	~8000 mg
**Υ-tocopherol**	21 mg
**Sodium**	2 mg
**Potassium**	441 mg
**Magnesium**	158 mg
**Calcium**	98 mg
**Phytomelatonin**	350 ng

**Table 2 foods-10-01231-t002:** Comparative data on plasmatic redox biomarkers according to the adherence to MedDiet.

	Type 2 Diabetics with DR		Type 2 Diabetics without DR		*p* Value
	*Poor Adherence MedDiet*	*Good Adherence MedDiet*	*Poor Adherence MedDiet*	*Good Adherence MedDiet*	
**MDA/TBARS (mm/L)**	4 ± 1	3 ± 1	2 ± 1	2 ± 1	*p* < 0.001
**TAC** **(mM)**	2 ± 1	2 ± 2	3 ± 1	3 ± 2	*p* < 0.051

DR: Diabetic Retinopathy; MDA/TBARS: Malondialdehyde/Thiobarbituric Acid Reactive Substances; TAC: Total Antioxidant Capacity; MedDiet: Mediterranean Diet. Data are shown as mean ± standard deviation. *p* value obtained from ANOVA analysis [40].

**Table 3 foods-10-01231-t003:** Adherence to MedDiet in eye patients and healthy controls [155].

	CG (*n* = 58)	Ocular Pathologies (*n* = 52)		
		Cataracts (*n* = 16)	Glaucoma (*n* = 13)	DR (*n* = 23)
**Adherence level**	8.4 ± 1.8	8.8 ± 2.3	8.9 ± 2.2	8.3 ± 2.1
***p* value**	Ref	0.519	0.371	0.775

CG: control group; DR: diabetic retinopathy.

**Table 4 foods-10-01231-t004:** Most relevant studies (alphabetically sorted by nutritional supplementation type) in diabetic retinopathy, glaucoma and ocular surface disorders/dry eye disease.

**Diabetic Retinopathy**					
**Oral Supplementation**	**Year**	**Authors**	**N**	**Follow-Up**	**Results of Intervention**
Complex formula [DiVFuSS formula]	2016	Chous et al. [167]	67	6 months	Improvement in visual function without macular thickness change.
Complex formula [Vitalux Forte^®^]	2011	Garcia-Medina et al. [165]	105	5 years	Visual acuity unchanged. Slower progression of treated group.
Lutein and zeaxanthin	2011	Hu et al. [166]	90	3 months	Better visual acuity, contrast sensitivity and decrease of foveal thickness.
Vitamin E	1999	Bursell et al. [164]	45	8 months	Improvement of retinal blood flow after supplementation.
Antioxidants, Carotenoids, Trace metals and Omega 3 Fatty Acids [Nutrof Omega ^®^ formula]	2015	Roig-Revert et al., VSDR group [40]	360	18 months	Decreased plasmatic oxidative level and increased antioxidant activity was seen in T2DM patients.
Antioxidants, Carotenoids, Trace metals and Omega 3 Fatty Acids [Nutrof Omega ^®^ formula]	2020	Sanz-González et al. VSDR group [154]	575	38 months	Reduced oxidative load and dietary prophylaxis/adjunctive intervention for patients at risk of diabetic retinopathy.
**Glaucoma**					
**Oral Supplementation**	**Year**	**Authors**	**N**	**Follow-Up**	**Results of Intervention**
AREDS-based formulas	2015	Garcia-Medina et al. [171]	117	2 years	No differences in visual field indexes, RGCl complex, peripapillary retinal nerve fiber layer.
Black currant anthocyanins	2012	Ohguro et al. [168]	38	24 months	Better mean deviation change (visual field index) in supplemented group.
Black currant anthocyanins	2013	Ohguro et al. [169]	21	4 weeks	IOP decrease at 2 and 4 weeks in treated group but no change in placebo group.
Docosahexaenoic acid	2018	Romeo Villadoniga et al. [173]	47	6 months	IOP decrease at 3 and 6 months in treated eyes.
Formula containing forskolin, homotaurine, carnosine, and folic acid	2016	Mutolo et al. [172]	44	1 year	IOP lowering, ERG improvement and foveal sensitivity.
Antioxidants and Omega 3 fatty acids [Brudysec ^®^ formula]	2013	Galbis-Estrada et al. [45]	97	3 months	Reduced inflammation biomarkers in glaucomatous tears.
Ginkgo biloba extract	2013	Lee et al. [170]	42	12 years	Slower progression of visual field damage in treated patients.
**Ocular Surface Disorders/Dry Eyes**					
Omega 3 and Omega 6 polyunsaturated fatty acids	2019	Downie et al. [174]		4214	Cochrane systematic review of clinical trials evidence uncertain/inconsistent for DEDs.
Omega 3 and Omega 6 polyunsaturated fatty acids	2014	Liu and Ji, [175]		790	Meta-analyses of randomized, placebo-controlled studies. Improvement of clinical tests, individual-reported symptoms in DEDs.
Antioxidants and Omega 3 fatty acids [Brudysec ^®^ formula]	2013	Pinazo-Duran et al. [44]		66	Benefit DED patients. Improvement of dry eye-related quality of life. Ameliorating clinical tests and tear expression of inflammatory mediators.
Antioxidants and Omega 3 fatty acids [Brudysec ^®^ formula]	2015	Galbis-Estrada et al. [123]		90	Improvement of subjective dry-eye symptoms by changing the tear metabolomic profile
Omega 3 fatty acids [Brudy Derm Dry Eye Gel ^®^]	2021	Pinazo-Durán et al. [177]		72	Ameliorating ocular surface relief and decreasing cytokine expression in tears from contact lenses users.

## Data Availability

Data from this study is contained within this article.

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
