# Peer review of "Searching for the Antioxidant, Anti-Inflammatory, and Neuroprotective Potential of Natural Food and Nutritional Supplements for Ocular Health in the Mediterranean Population"

_foods, 2021, doi:10.3390/foods10061231_

Round 1

Reviewer 1 Report

The authors present a work entitled " Searching for the antioxidant, anti-inflammatory, and neuroprotective potential of natural food and nutritional supplements for ocular health in the mediterranean population". The manuscript deals with a subject of significant interest and can certainly be fascinating to readers. However, in my opinion, this review article needs to undergo major revision. First of all, it is necessary to highlight the objectives of the work and the moments of novelty of the review. Here are the suggestions / corrections to improve the quality of the manuscript.Throughout the text there are numerous translation and grammar errors. Please, the text should be checked by a native English speaker

Is there a precise reason for choosing the botanical species treated (broccoli, saffron, tiger nuts and walnuts)?

Please, rduce the number of keywords in accordance with Instructions for Authors

Ln 61: pleas, correct key word by keywords

Ln 72: Please, correct dissability by disability

Ln 72, 80: Please, correct celular by cellular throughout the text

Ln 85: Please, correct hydroxil by hydroxyl

Ln 87: efectors?

Ln 88-90: “Nonetheless, ROS can undergo a window swichting physiological and biological activity, being molecules that infringe damage (as occurring in oxidative stress) and/or regulating signaling pathways”, please, rewrite this sentence more clearly, it’s relevant to explain the balance antioxidants/prooxidants with the relative effects (phisiological/pathological)

Ln 91: “ROS/RNS” or “ROS and RNS”, in the first case it means a ratio

Ln 95: B lymphocytes help or helper?

Ln 106: under under

Ln 109:, please, replace ROS/RNS with ROS and RNS

Ln 167: please, replace “by products” with “by-products”

Ln : “0.5. d.u. like 175 mild levels and 0.5 as high levels, maybe you mean more than 0.5 in the last sentence?

Ln 180-185: please rewrite and organize this long sentence more clearly and fluently by inserting the right punctuation

Ln 227: “This study mainly suggest that the broccoli course improved…”, please, improve the expression “the broccoli course”

Ln 239: “crocin isomers, carotenoids (zeaxanthin) antioxidants (lycopene)…”, I did not understand this statement: crocin and crocetin are also carotenoids, zeaxanthin is an oxygenated carotenoid and lycopene a non-oxygenated carotenoid with high antioxidant power, but generally all the others also have this property. Please rewrite this sentence taking these concepts into account.

Ln 443: age-related macular degeneration (AMD), please use only the acronym as it is already explained above

Ln 455: oxilipines, do you mean oxylipins?

Ln 488: please, correct the double point after ref [146]

Ln 510: “diabetics with or withour DR were analyzed” correct by diabetics with or without DR were analyzed

Ln 529: report anàlisis in english

Ln 669: please, correct Clossing remarks by Concluding remarks

Section 5: please, add a few sentences in the conclusions to highlight the innovative contribution of this review and therefore its usefulness

Table 1: In my opinion, tab 1 is set incorrectly: it is not clear whether the properties are ascribed to one or more or all the substances listed in column 1. Please reconstruct this table so that the information is clear and not confusing

Author Response

ANSWERS TO REVIEWER 1

The authors present a work entitled " Searching for the antioxidant, anti-inflammatory, and neuroprotective potential of natural food and nutritional supplements for ocular health in the mediterranean population". The manuscript deals with a subject of significant interest and can certainly be fascinating to readers. However, in my opinion, this review article needs to undergo major revision. First of all, it is necessary to highlight the objectives of the work and the moments of novelty of the review. Here are the suggestions / corrections to improve the quality of the manuscript. Throughout the text there are numerous translation and grammar errors. Please, the text should be checked by a native English speaker

We are very grateful to the Reviewer for pointing to this issue, and as the Reviewer suggest us, we have highlighted the objectives of the work in introduction section as well as the novelty of the review.

Lines 140-146: “In fact, our review article included four subsections regarding the benefits of natural food for vision health. Broccoli, nuts, safron and tigernuts are awesome single foods that can help preventing/managing ocular diseases and also can help fighting against certain risk factors-related to visual impairment. Practically all of these display anti-inflammatory, detoxicating, anti-angiogenic, anti-apoptotic, photoprotective, antioxidant and neuroprotective effects to some extent. Because of this, here we sought to address the role of diet and nutrition in the eyes and vision, focusing on the potential benefit of natural food (broccoli, saffron, tiger nuts and walnuts), the MedDiet and nutraceutical supplements on eye health as well as in preventing vision loss.”

Novelty is related to the broccoli and chufa de Valencia studies, as well as the revision of Mediterranean diet, nutraceutical supplements and natural food related to ocular diseases.

We have also given the MS to a native English speaker to improve the readable for readers.

Is there a precise reason for choosing the botanical species treated (broccoli, saffron, tiger nuts and walnuts)?

Regarding to this topic, we want to express that all-natural food we have chosen are enriched in antioxidants and anti-inflammatory compounds such as carotenes, vitamins, and fatty acids, making their intake potentially beneficial for eye health.

Please, rduce the number of keywords in accordance with Instructions for Authors

We are very grateful to the Reviewer for pointing to this issue. We have rewritten the keywords to maintain the number of 10. And we have included three properties as one keyword: Antioxidant, Anti-inflammatory and Neuroprotective properties

Ln 61: pleas, correct key word by keywords

Thanks to the reviewer for this called of attention. It has been corrected.

Ln 72: Please, correct dissability by disability

Thank you very much for notice us this error. Now it is corrected in the new MS version.

Ln 72, 80: Please, correct celular by cellular throughout the text

Thank you very much for notice us this error. Now it is corrected in the new MS version.

Ln 85: Please, correct hydroxil by hydroxyl

Thank you very much for notice us this involuntary error It has been corrected along the MS.

Ln 87: efectors?

Thank you very much for notice us this error. We have corrected the misspelling as effectors.

Ln 88-90: “Nonetheless, ROS can undergo a window swichting physiological and biological activity, being molecules that infringe damage (as occurring in oxidative stress) and/or regulating signaling pathways”, please, rewrite this sentence more clearly, it’s relevant to explain the balance antioxidants/prooxidants with the relative effects (phisiological/pathological)

We are very grateful to the Reviewer for pointing to this issue. We have changed the sentence for one new. “Active ROS present a dual role, acting as both destructive and constructive species. Thus, they participate in many activities for the preservation of cellular homeostasis, but in high concentrations they lead to a situation of oxidative stress involved in the damage of cellular structures.” Now in lines 94-97 on page 3.

Ln 91: “ROS/RNS” or “ROS and RNS”, in the first case it means a ratio

Thanks to the reviewer for this called of attention. We have revised all the MS and corrected. To maintain the nomenclature, we have also changed the sentence where we made the abbreviation to clarifier the manuscript as: “Dysbalance between the generation of reactive oxygen and nitrogen species (ROS and RNS)” on line 88. This new form has been adopted along the MS.

Ln 95: B lymphocytes help or helper?

Thank you very much for notice us this involuntary error. We have corrected.

Ln 106: under under

Thank you very much for notice us this involuntary error. One under has been eliminated.

Ln 109:, please, replace ROS/RNS with ROS and RNS

This error has been corrected along the MS.

Ln 167: please, replace “by products” with “by-products”

Thanks to the reviewer for this called of attention. It has been corrected.

Ln : “0.5. d.u. like 175 mild levels and 0.5 as high levels, maybe you mean more than 0.5 in the last sentence?

Regarding to this topic, we want to express our grateful to the Reviewer. We have changed the sentence to a more understable form. “below 0.2 d.u. should be taken as low levels, 0.20.5. d.u. as mild levels and more than 0.5 as high levels”

Ln 180-185: please rewrite and organize this long sentence more clearly and fluently by inserting the right punctuation

We agree completely with the referee, and are very grateful for the opportunity to clarify our ms. In the new version we have changed this sentence as: “We determined plasma total antioxidant capacity (TAC), by enzymatic-colorimetric assays, as previously published [6-11] and MPOD determined from retinographies from the right eye (RE) and left eye (LE) collected with the VISUCAM 500 ® (Carl Zeiss Meditec Iberia, Tres cantos, Madrid, Spain)”

¡Ln 227: “This study mainly suggest that the broccoli course improved…”, please, improve the expression “the broccoli course”

Corrected. We have changed “course” for “intake”.

Ln 239: “crocin isomers, carotenoids (zeaxanthin) antioxidants (lycopene)…”, I did not understand this statement: crocin and crocetin are also carotenoids, zeaxanthin is an oxygenated carotenoid and lycopene a non-oxygenated carotenoid with high antioxidant power, but generally all the others also have this property. Please rewrite this sentence taking these concepts into account.

We agree with the reviewer's comments. We have made the changes according to the reviewers' suggestions and they have been included in the new version of the manuscript.

Ln 443: age-related macular degeneration (AMD), please use only the acronym as it is already explained above

Authors are very grateful to the Reviewer for notice us this error. We have corrected and used the abbreviation along the MS. This pinpointed has notice us the use of Type 2 Diabetes mellitus as T2DM, and Mediterranean Diet as MedDiet. We have corrected along the MS.

Ln 455: oxilipines, do you mean oxylipins?

We agree with the Reviewer's comments. It has been corrected.

Ln 488: please, correct the double point after ref [146]

We agree with the Reviewer's comments. It has been corrected.

Ln 510: “diabetics with or withour DR were analyzed” correct by diabetics with or without DR were analyzed

Thank you very much for notice us this involuntary error. It has been corrected.

Ln 529: report anàlisis in English

Thank you very much for notice us this involuntary error. It has been corrected.

Ln 669: please, correct Clossing remarks by Concluding remarks

We agree with the reviewer's comments. It has been corrected.

Section 5: please, add a few sentences in the conclusions to highlight the innovative contribution of this review and therefore its usefulness

We agree with the reviewer's comments. Noe the concluding remarks are in section 7. As the Reviewer suggest us, we have included a new paragraph: “In this review we clearly demonstrates that natural food is essential for the eyes and vision and these discoveries are common to different countries where the revised studies were conducted, but also important differences have been detected across the revised studies. With this basis we may propose that neglecting the potential of natural food, the benefits of the MedDiet and the positive effects of appropriated nutritional supplements may result in high costs for the societies worldwide regarding the ocular health”.

Table 1: In my opinion, tab 1 is set incorrectly: it is not clear whether the properties are ascribed to one or more or all the substances listed in column 1. Please reconstruct this table so that the information is clear and not confusing

Thank you very much for notice us this confusing aspect. We have changed the table, explaining the general properties along the ms. Lines 469-471 on page 12.

We thank the Reviewer for the time used for this revision, and we are extremely grateful to the Reviewer for positive criticism to improve our work. Related to this, we have revised the MS writing the abbreviations along the MS, and corrected misspelling words. Even more, some new cites have been added thorough the MS.

Reviewer 2 Report

The topic of manuscript is very interesting and of great importance for the further scientific researches. The manuscript is well organized. This review work was designed and carried out by a multidisciplinary group involved in ophthalmology and ophthalmic research.

The proposed work is a good summary of the results of research in this subject.

As a reviewer I have few suggestion:

  1. Abstract is well written and adequately presents the aim and the basic results of the study. Nevertheless, authors should add 2-3 lines about the basic methods used (eg number and type of studies reviewed).
  2. The sentence “ Nutrition clearly makes a difference to the eye health and vision care. In fact, coordinated, multidisciplinary 133 interventions are essential to deal with the role of natural food, and nutritional supplements to achieve a better 134 knowledge of the diet and ocular diseases, as in the present review” doesn't make sense. This section should be corrected to make it clear what the purpose of the work was. The aim of the work must be a development of the goal described in the abstract.
  3. Methods section is necessary to complete data. Specifically, how many studies reviewed, the type of the studies, the databases used (eg scopus etc), the inclusion criteria of the type of each study and the kind of review (Is this a systematic review?)
  4. Figure 1, Figure 4 and Figure 5 should be cited in the text.
  5. The final conclusions should be redrafted pointing to the specific information that has been discussed in the text. The present form of this chapter is too general and imprecise. You should refer to both food and supplements.

Author Response

The topic of manuscript is very interesting and of great importance for the further scientific researches. The manuscript is well organized. This review work was designed and carried out by a multidisciplinary group involved in ophthalmology and ophthalmic research.

The proposed work is a good summary of the results of research in this subject.

As a reviewer I have few suggestion:

We extremely thank the Reviewer for calling our attention to all these points we have explained one by one.

  • Abstract is well written and adequately presents the aim and the basic results of the study. Nevertheless, authors should add 2-3 lines about the basic methods used (eg number and type of studies reviewed).

We thank the Reviewer to point us this comment, and as the Reviewer suggest us, we have added this new paragraph: “A systematic search of PubMed, Web of Science,  hand-searched publications and historical archives were perfomed by the professionals invoved in this study, to include peer-reviewed articles in which natural food, nutrient content, and  its potential relationship with ocular health. Five ophthalmologist and two researchers collected the characteristics, quality and suitability of the above studies. Finally 179 papers from 1983 to 2021 were enclosed, mainly related to natural food, Mediterranean diet (MedDiet) and nutraceutic supplementation. For the first time original studies with broccoli and tigernut (chufa de Valencia) regarding the ocular surface dysfunction, macular degeneration, diabetic retinopathy and glaucoma were enclosed.” Lines 54-59.

  • The sentence “ Nutrition clearly makes a difference to the eye health and vision care. In fact, coordinated, multidisciplinary 133 interventions are essential to deal with the role of natural food, and nutritional supplements to achieve a better 134 knowledge of the diet and ocular diseases, as in the present review” doesn't make sense. This section should be corrected to make it clear what the purpose of the work was. The aim of the work must be a development of the goal described in the abstract.

We agree completely with the referee, and are very grateful for the opportunity to improve our MS. To this end, we have included a new cite, number 54, and also the following paragraph: “In fact, our review article included four subsections regarding the benefits of natural food for vision health. Broccoli, nuts, safron and tigernuts are awesome single foods that can help preventing/managing ocular diseases and also can help fighting against certain risk factors-related to visual impairment. Practically all of these display anti-inflammatory, detoxicating, anti-angiogenic, anti-apoptotic, photoprotective, antioxidant and neuroprotective effects to some extent. Because of this, here we sought to address the role of diet and nutrition in the eyes and vision, focusing on the potential benefit of natural food (broccoli, saffron, tiger nuts and walnuts), the MedDiet and nutraceutical supplements on eye health as well as in preventing vision loss.” Lines 142-148.

  • Methods section is necessary to complete data. Specifically, how many studies reviewed, the type of the studies, the databases used (eg scopus etc), the inclusion criteria of the type of each study and the kind of review (Is this a systematic review?)

We are very grateful to the Reviewer for pointing to this issue. As the Reviewer suggest we have included a new section related to Material and Methods on page 4. We have also included three cites in this new section (59-61).

  •  Figure 1, Figure 4 and Figure 5 should be cited in the text.

Regarding to this topic, we want to express our grateful to the Reviewer. We have revised the Figures citation along the text, as well as the Figures. Some of them have been changed for a new version to improve the MS. We have substituted the old Figure 5 for a new one being more important for readable the MS.

  • The final conclusions should be redrafted pointing to the specific information that has been discussed in the text. The present form of this chapter is too general and imprecise. You should refer to both food and supplements.

We are very grateful to the Reviewer for pointing to this issue. As the Reviewer suggest us, we have included a new paragraph: “In this review we clearly demonstrate that natural food is essential for the eyes and vision and these discoveries are common to different countries where the revised studies were conducted, but also important differences have been detected across the revised studies. With this basis we may propose that neglecting the potential of natural food, the benefits of the MedDiet and the positive effects of appropriated nutritional supplements may result in high costs for the societies worldwide regarding the ocular health” in new section 7 on page 17.

We are extremely grateful to the Reviewer for outstanding help and positive criticism to improve our work. We thank the Reviewer for the time used for this revision. We have revised the MS writing the abbreviations along the MS, and corrected misspelling words. Some new cites have been added thorough the MS.

Round 2

Reviewer 1 Report

I am grateful to the authors for improving the manuscript by accepting all suggestions from reviewers. I believe that in the current version the manuscript is worthy of publication

Reviewer 2 Report

I accept the content of the manuscript after the changes made in accordance with the recommendations in the review. I recommend that the manuscript should be accepted in the present form